# PHASE TRANSITION FOR DETECTING A SMALL COMMUNITY IN A LARGE NETWORK

**Jiashun Jin**
Carnegie Mellon University
jiashun@stat.cmu.edu

**Zheng Tracy Ke**
Harvard University
zke@fas.harvard.edu

**Paxton Turner**
Harvard University
paxtonturner@fas.harvard.edu

**Anru R. Zhang**
Duke University
anru.zhang@duke.edu

## ABSTRACT

How to detect a small community in a large network is an interesting problem, including clique detection as a special case, where a naive degree-based $\chi^2$-test was shown to be powerful in the presence of an Erdős-Renyi background. Using Sinkhorn's theorem, we show that the signal captured by the $\chi^2$-test may be a modeling artifact, and it may disappear once we replace the Erdős-Renyi model by a broader network model. We show that the recent SgnQ test is more appropriate for such a setting. The test is optimal in detecting communities with sizes comparable to the whole network, but has never been studied for our setting, which is substantially different and more challenging. Using a degree-corrected block model (DCBM), we establish phase transitions of this testing problem concerning the size of the small community and the edge densities in small and large communities. When the size of the small community is larger than $\sqrt{n}$, the SgnQ test is optimal for it attains the computational lower bound (CLB), the information lower bound for methods allowing polynomial computation time. When the size of the small community is smaller than $\sqrt{n}$, we establish the parameter regime where the SgnQ test has full power and make some conjectures of the CLB. We also study the classical information lower bound (LB) and show that there is always a gap between the CLB and LB in our range of interest.

## 1 INTRODUCTION

Consider an undirected network with $n$ nodes and $K$ communities. We assume $n$ is large and the network is connected for convenience. We are interested in testing whether $K = 1$ or $K > 1$ and the sizes of some of the communities are much smaller than $n$ (communities are scientifically meaningful but mathematically hard to define; intuitively, they are clusters of nodes that have more edges "within" than "across" (Jin, 2015; Zhao et al., 2012)). The problem is a special case of network global testing, a topic that has received a lot of attention (e.g., Jin et al. (2018; 2021b)). However, existing works focused on the so-called *balanced case*, where the sizes of communities are at the same order. Our case is *severely unbalanced*, where the sizes of some communities are much smaller than $n$ (e.g., $n^\varepsilon$).

The problem also includes clique detection (a problem of primary interest in graph learning (Alon et al., 1998; Ron & Feige, 2010)) as a special case. Along this line, Arias-Castro & Verzelen (2014); Verzelen & Arias-Castro (2015) have made remarkable progress. In detail, they considered the problem of testing whether a graph is generated from a one-parameter Erdős-Renyi model or a two-parameter model: for any nodes $1 \leq i, j \leq n$, the probability that they have an edge equals $b$ if $i, j$ both are in a small planted subset and equals $a$ otherwise. A remarkable conclusion of these papers is: a naive degree-based $\chi^2$-test is optimal, provided that the clique size is in a certain range. Therefore, at first glance, it seems that the problem has been elegantly solved, at least to some extent.

Unfortunately, recent progress in network testing tells a very different story: the signal captured by the $\chi^2$-test may be a modeling artifact. It may disappear once we replace the models in Arias-Castro

& Verzelen (2014); Verzelen & Arias-Castro (2015) by a properly broader model. When this happens, the $\chi^2$-test will be asymptotically powerless in the whole range of parameter space.

We explain the idea with the popular *Degree-Corrected Block Model (DCBM)* (Karrer & Newman, 2011), though it is valid in broader settings. Let $A \in \mathbb{R}^{n,n}$ be the network adjacency matrix, where $A(i,j) \in \{0,1\}$ indicates whether there is an edge between nodes $i$ and $j$, $1 \le i, j \le n$. By convention, we do not allow for self-edges, so the diagonals of $A$ are always $0$. Suppose there are $K$ communities, $\mathcal{C}_1, \ldots, \mathcal{C}_K$. For each node $i$, $1 \le i \le n$, we use a parameter $\theta_i$ to model the degree heterogeneity and $\pi_i$ to model the membership: when $i \in \mathcal{C}_k$, $\pi_i(\ell) = 1$ if $\ell = k$ and $\pi_i(\ell) = 0$ otherwise. For a $K \times K$ symmetric and irreducible non-negative matrix $P$ that models the community structure, DCBM assumes that the upper triangle of $A$ contains independent Bernoulli random variables satisfying[1]

$$\mathbb{P}(A(i,j) = 1) = \theta_i \theta_j \pi_i' P \pi_j, \qquad 1 \le i, j \le n. \tag{1.1}$$

In practice, we interpret $P(k, \ell)$ as the baseline connecting probability between communities $k$ and $\ell$. Write $\theta = (\theta_1, \theta_2, \ldots, \theta_n)'$, $\Pi = [\pi_1, \pi_2, \ldots, \pi_n]'$, and $\Theta = \mathrm{diag}(\theta) \equiv \mathrm{diag}(\theta_1, \theta_2, \ldots, \theta_n)$. Introduce $n \times n$ matrices $\Omega$ and $W$ by $\Omega = \Theta \Pi P \Pi' \Theta$ and $W = A - \mathbb{E}[A]$. We can re-write (1.1) as

$$A = \Omega - \mathrm{diag}(\Omega) + W. \tag{1.2}$$

We call $\Omega$ the *Bernoulli probability matrix* and $W$ the noise matrix. When $\theta_i$ in the same community are equal, DCBM reduces to the Stochastic Block Model (SBM) (Holland et al., 1983). When $K = 1$, the SBM reduces to the Erdős-Renyi model, where $\Omega(i,j)$ take the same value for all $1 \le i, j \le n$.

We first describe why the signal captured by the $\chi^2$-test in Arias-Castro & Verzelen (2014); Verzelen & Arias-Castro (2015) is a modeling artifact. Using Sinkhorn's matrix scaling theorem (Sinkhorn, 1974), it is possible to build a null DCBM with $K = 1$ that has no community structure and an alternative DCBM with $K \ge 2$ and clear community structure such that the two models have the *same* expected degrees. Thus, we do not expect that degree-based test such as $\chi^2$ can tell them apart. We make this Sinkhorn argument precise in Section 2.1 and show the failure of $\chi^2$ in Theorem 2.3.

In the Erdős-Renyi setting in Arias-Castro & Verzelen (2014), the null has one parameter and the alternative has two parameters. In such a setting, we cannot have degree-matching. In these cases, a naive degree-based $\chi^2$-test may have good power, but it is due to the very specific models they choose. For clique detection in more realistic settings, we prefer to use a broader model such as the DCBM, where by the degree-matching argument above, the $\chi^2$-test is asymptotically powerless.

This motivates us to look for a different test. One candidate is the scan statistic Bogerd et al. (2021). However, a scan statistic is only computationally feasible when each time we scan a very small subset of nodes. For example, if each time we only scan a finite number of nodes, then the computational cost is polynomial; we call the test the *Economic Scan Test (EST)*. Another candidate may come from the Signed-Polygon test family (Jin et al., 2021b), including the Signed-Quadrilateral (SgnQ) as a special case. Let $\hat{\eta} = (\mathbf{1}_n A \mathbf{1}_n)^{-1/2} A \mathbf{1}_n$ and $\widehat{A} = A - \hat{\eta}\hat{\eta}$. Define $Q_n = \sum_{i_1, i_2, i_3, i_4 (dist)} \widehat{A}_{i_1 i_2} \widehat{A}_{i_2 i_3} \widehat{A}_{i_3 i_4} \widehat{A}_{i_4 i_1}$ where the shorthand $(dist)$ indicates we sum over distinct indices. The SgnQ test statistic is

$$\psi_n = \left[ Q_n - 2(\|\hat{\eta}\|^2 - 1)^2 \right] / \sqrt{8(\|\hat{\eta}\|^2 - 1)^4}. \tag{1.3}$$

SgnQ is computationally attractive because it can be evaluated in time $O(n^2 \bar{d})$, where $\bar{d}$ is the average degree of the network (Jin et al., 2021b).

Moreover, it was shown in Jin et al. (2021b) that (a) when $K = 1$ (the null case), $\psi_n \to N(0,1)$, and (b) when $K > 1$ and all communities are at the same order (i.e., a balanced alternative case), the SgnQ test achieves the classical information lower bound (LB) for global testing and so is optimal. Unfortunately, our case is much more delicate: the signal of interest is contained in a community with a size that is much smaller than $n$ (e.g., $n^\varepsilon$), so the signal can be easily overshadowed by the noise term of $Q_n$. Even in the simple alternative case where we only have two communities (with sizes $N$ and $(n - N)$), it is unclear (a) how the lower bounds vary as $N/n \to 0$, and especially whether there is a gap between the computation lower bound (CLB) and classical information lower bound (LB), and (b) to what extent the SgnQ test attains the CLB and so is optimal.

---

[1]In this work we use $M'$ to denote the transpose of a matrix or vector $M$.

## 1.1 RESULTS AND CONTRIBUTIONS

We consider the problem of detecting a small community in the DCBM. In this work, we specifically focus on the case $K = 2$ as this problem already displays a rich set of phase transitions, and we believe it captures the essential behavior for constant $K > 1$. Let $N \ll n$ denote the size of this small community under the alternative. Our first contribution analyzes the power of SgnQ for this problem, extending results of Jin et al. (2021b) that focus on the balanced case. Let $\lambda_1 = \lambda_1(\Omega)$. In Section 2.2, we define a population counterpart $\tilde{\Omega}$ of $\hat{A}$ and let $\tilde{\lambda} = \lambda_1(\tilde{\Omega})$. We show that SgnQ has full power if $\tilde{\lambda}_1/\sqrt{\lambda_2} \to \infty$, which reduces to $N(a-c)/\sqrt{nc} \to \infty$ in the SBM case.

For optimality, we obtain a computational lower bound (CLB), relying on the low-degree polynomial conjecture, which is a standard approach in studying CLB (e.g., Kunisky et al. (2019)). Consider a case where $K = 2$ and we have a small community with size $N$. Suppose the edge probability within the community and outside the community are $a$ and $c$, where $a > c$. The quantity $(a-c)/\sqrt{c}$ acts as the *Node-wise Signal-to-Noise Ratio (SNR)* for the detection problem.[2] When $N \gg \sqrt{n}$, we find that the CLB is completely determined by $N$ and node-wise SNR; moreover, SgnQ matches with the CLB and is optimal. When $N \ll \sqrt{n}$, the situation is more subtle: if the node-wise SNR $(a-c)/\sqrt{c} \to 0$ (weak signal case), we show the problem is computationally hard and the LB depends on $N$ and the node-wise SNR. If $(a-c)/\sqrt{c} \gg n^{1/2}$ (strong signal case), then SgnQ solves the detection problem. In the range $1 \ll (a-c)/\sqrt{c} \ll n^{1/2}$ (moderate signal case), the CLB depends on not only $N$ and the node-wise SNR but also the background edge density $c$. In this regime, we make conjectures of the CLB, from the study of the aforementioned economic scan test (EST). Our results are summarized in Figure 1 and explained in full detail in Section 2.7.

We also obtain the classical information lower bound (LB), and discover that as $N/n \to 0$, there is big gap between CLB and LB. Notably the LB is achieved by an (inefficient) signed scan test. In the balanced case in Jin et al. (2021b), the SgnQ test is optimal among all tests (even those that are allowed unbounded computation time), and such a gap does not exist.

We also show that that the naive degree-based $\chi^2$-test is asymptotically powerless due to the aforementioned degree-matching phenomenon.

Our statistical lower bound, computational lower bound, and the powerlessness of $\chi^2$ based on degree-matching are also valid for all $K > 2$ since any model with $K \geq 2$ contains $K = 2$ as a special case. We also expect that our lower bounds are tight for these broader models and that our lower bound constructions for $K = 2$ represent the least favorable cases when community sizes are severely unbalanced.

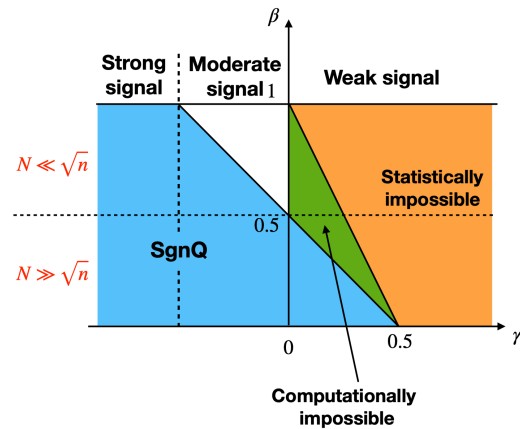

Figure 1: Phase diagram ($(a-c)/\sqrt{c} = n^{-\gamma}$ and $N = n^{1-\beta}$).

Compared to Verzelen & Arias-Castro (2015); Arias-Castro & Verzelen (2014), we consider network global testing in a more realistic setting, and show that optimal tests there (i.e., a naive degree-based $\chi^2$ test) may be asymptotically powerless here. Compared with Bogerd et al. (2021), our setting is very different (they considered a setting where both the null and alternative are DCBM with $K = 1$). Compared to the study in the balanced case (e.g., Jin et al. (2018; 2021b); Gao & Lafferty (2017)), our study is more challenging for two reasons. First, in the balanced case, there is no gap between the UB (the upper bound provided by the SgnQ test) and LB, so there is no need to derive the CLB, which is usually technical demanding. Second, the size of the smaller community can get as small as $n^\varepsilon$, where $\varepsilon > 0$ is any constant. Due this imbalance in community sizes, the techniques of Jin et al. (2021b) do not directly apply. As a result, our proof involves the careful study of the 256 terms that compose SgnQ, which requires using bounds tailored specifically for the severely unbalanced case.

---

[2]Note that the node-wise SNR captures the ratio of the mean difference and standard deviation of Bernoulli($a$) versus Bernoulli($c$), which motivates our terminology.

Our study of the CLB is connected to that of Hajek et al. (2015) in the Erdös-Renyi setting of Arias-Castro & Verzelen (2014). Hajek et al. (2015) proved via computational reducibility that the naive $\chi^2$-test is the optimal polynomial-time test (conditionally on the planted clique hypothesis). We also note work of Chen & Xu (2016) that studied a $K$-cluster generalization of the Erdös-Renyi model of Arias-Castro & Verzelen (2014); Verzelen & Arias-Castro (2015) and provided conjectures of the CLB. Compared to our setting, these models are very different because the expected degree profiles of the null and alternative differ significantly. In this work we consider the DCBM model, where due to the subtle phenomenon of degree matching between the null and alternative hypotheses, both CLB and LB are different from those obtained by Hajek et al. (2015).

**Notations:** We use $\mathbf{1}_n$ to denote a $n$-dimensional vector of ones. For a vector $\theta = (\theta_1, \ldots, \theta_n)$, $\mathrm{diag}(\theta)$ is the diagonal matrix where the $i$-th diagonal entry is $\theta_i$. For a matrix $\Omega \in \mathbb{R}^{n \times n}$, $\mathrm{diag}(\Omega)$ is the diagonal matrix where the $i$-th diagonal entry is $\Omega(i,i)$. For a vector $\theta \in \mathbb{R}^n$, $\theta_{max} = \max\{\theta_1, \ldots, \theta_n\}$ and $\theta_{min} = \min\{\theta_1, \ldots, \theta_n\}$. For two positive sequences $\{a_n\}$ and $\{b_n\}$, we write $a_n \asymp b_n$ if $c_1 \leq a_n/b_n \leq c_2$ for constants $c_2 > c_1 > 0$. We say $a_n \sim b_n$ if $(a_n/b_n) = 1 + o(1)$.

## 2 MAIN RESULTS

In Section 2.1, following our discussion on Sinkhorn's theorem in Section 1, we introduce calibrations (including conditions on identifiability and balance) that are appropriate for severely unbalanced DCBM and illustrate with some examples. In Sections 2.2-2.3, we analyze the power of the SgnQ test and compare it with the $\chi^2$-test. In Sections 2.4-2.5, we discuss the information lower bounds (both the LB and CLB) and show that SgnQ test is optimal among polynomial time tests, when $N \gg \sqrt{n}$. In Section 2.6, we study the EST and make some conjectures of the CLB when $N \ll \sqrt{n}$. In Section 2.7, we summarize our results and present the phase transitions.

### 2.1 DCBM FOR SEVERELY UNBALANCED NETWORKS: IDENTIFIABILITY, BALANCE METRICS, AND GLOBAL TESTING

In the DCBM (1.1)-(1.2), $\Omega = \Theta \Pi P \Pi' \Theta$. It is known that the matrices $(\Theta, \Pi, P)$ are not identifiable. One issue is that $(\Pi, P)$ are only unique up to a permutation: for a $K \times K$ permutation matrix $Q$, $\Pi P \Pi = (\Pi Q)(Q'PQ)(\Pi Q)'$. This issue is easily fixable in applications so is usually neglected. A bigger issue is that, $(\Theta, P)$ are not uniquely defined. For example, fixing a positive diagonal matrix $D \in \mathbb{R}^{K \times K}$, let $P^* = DPD$ and $\Theta^* = \mathrm{diag}(\theta_1^*, \theta_2^*, \ldots, \theta_n^*)$ where $\theta_i^* = \theta_i / \sqrt{D(k,k)}$ if $i \in \mathcal{C}_k$, $1 \leq k \leq K$. It is seen that $\Theta \Pi P \Pi' \Theta = \Theta^* \Pi P^* \Pi' \Theta^*$, so $(\Theta, P)$ are not uniquely defined.

To motivate our identifiability condition, we formalize the degree-matching argument discussed in the introduction. Fix $(\theta, P)$ and let $h = (h_1, \ldots, h_K)'$ and $h_k > 0$ is the fraction of nodes in community $k$, $1 \leq k \leq K$. By the main result of Sinkhorn (1974), there is a unique positive diagonal matrix $D = \mathrm{diag}(d_1, \ldots, d_K)$ such that $DPDh = \mathbf{1}_K$. Consider a pair of two DCBM, a null with $K = 1$ and an alternative with $K > 1$, with parameters $\Omega = \Theta \mathbf{1}_n \mathbf{1}_n' \Theta \equiv \theta \theta'$ and $\Omega^*(i,j) = \theta_i^* \theta_j^* \pi_i' P \pi_j$ with $\theta_i^* = d_k \theta_i$ if $i \in \mathcal{C}_k$, $1 \leq k \leq K$, respectively. Direct calculation shows that node $i$ has the same expected degree under the null and alternative.

There are many ways to resolve the issue. For example, in the balanced case (e.g., Jin et al. (2021b; 2022)), we can resolve it by requiring that $P$ has unit diagonals. However, for our case, this is inappropriate. Recall that, in practice, $P(k, \ell)$ represents as the baseline connecting probability between community $k$ and $\ell$. If we forcefully rescale $P$ to have a unit diagonal here, both $(P, \Theta)$ lose their practical meanings.

Motivated by the degree-matching argument, we propose an identifiability condition that is more appropriate for the severely unbalanced DCBM. By our discussion in Section 1, for any DCBM with a Bernoulli probability matrix $\Omega$, we can always use Sinkhorn's theorem to define $(\Theta, P)$ (while $\Pi$ is unchanged) such that for the new $(\Theta, P)$, $\Theta = \Theta \Pi P \Pi' \Theta$ and $Ph \propto \mathbf{1}_K$, where $h = (h_1, \ldots, h_K)'$ and $h_k > 0$ is the fraction of nodes in community $k$, $1 \leq k \leq K$. This motivates the following identifiability condition (which is more appropriate for our case):

$$\|\theta\|_1 = n, \qquad Ph \propto \mathbf{1}_K, \quad \text{where } h_k \text{ is fraction of nodes in } \mathcal{C}_k, 1 \leq k \leq K. \qquad (2.1)$$

**Lemma 2.1.** *For any $\Omega$ that satisfies the DCBM (1.2) and has positive diagonal elements, we can always find $(\Theta, \Pi, P)$ such that $\Omega = \Theta \Pi P \Pi' \Theta$ and (2.1) holds. Also, any $(\Theta, P)$ that satisfy $\Omega = \Theta \Pi P \Pi' \Theta$ and (2.1) are unique.*

Moreover, for network balance, the following two vectors in $\mathbb{R}^K$ are natural metrics:

$$d = (\|\theta\|_1)^{-1} \Pi' \Theta \mathbf{1}_n, \qquad g = (\|\theta\|)^{-2} \Pi' \Theta^2 \Pi \mathbf{1}_K, \qquad (2.2)$$

In the balanced case (e.g., Jin et al. (2021b; 2022)), we usually assume the entries of $d$ and $g$ are at the same order. For our setting, this is not the case.

Next we introduce the null and alternative hypotheses that we consider. Under each hypothesis, we impose the identifiability condition (2.1).

**General null model for the DCBM.** When $K = 1$ and $h = 1$, $P$ is scalar (say, $P = \alpha$), and $\Omega = \alpha \theta \theta'$ satisfies $\|\theta\|_1 = n$ by (2.1). The expected total degree is $\alpha(\|\theta\|_1^2 - \|\theta\|^2) \sim \alpha \|\theta\|_1^2 = n^2 \alpha$ under mild conditions, so we view $\alpha$ as the parameter for network sparsity. In this model, $d = g = 1$.

**Alternative model for the DCBM**. We assume $K = 2$ and that the sizes of the two communities, $\mathcal{C}_0$ and $\mathcal{C}_1$, are $(n - N)$ and $N$, respectively. For some positive numbers $a, b, c$, we have

$$P = \begin{bmatrix} a & b \\ b & c \end{bmatrix}, \qquad \text{and} \qquad \Omega(i,j) = \begin{cases} \theta_i \theta_j \cdot a, & \text{if } i, j \in \mathcal{C}_1, \\ \theta_i \theta_j \cdot c, & \text{if } i, j \in \mathcal{C}_0, \\ \theta_i \theta_j \cdot b, & \text{otherwise.} \end{cases} \qquad (2.3)$$

In the classical clique detection problem (e.g., Bogerd et al. (2021)), $a$ and $c$ are the baseline probability where two nodes have an edge when both of them are *in* the clique and *outside* the clique, respectively. By (2.1), $a\epsilon + b(1 - \epsilon) = b\epsilon + c(1 - \epsilon)$ if we write $\epsilon = N/n$. Therefore,

$$b = (c(n - N) - aN)/(n - 2N). \qquad (2.4)$$

Note that this is the *direct result* of Sinkhorn's theorem and the parameter calibration we choose, not a condition we choose for technical convenience. Write $d = (d_0, d_1)'$ and $g = (g_0, g_1)'$. It is seen that $d_0 = 1 - d_1$, $g_0 = 1 - g_0$, $d_1 = \|\theta\|_1^{-1} \sum_{i \in \mathcal{C}_1} \theta_i$, and $g_1 = \|\theta\|^{-2} \sum_{i \in \mathcal{C}_1} \theta_i^2$. If all $\theta_i$ are at the same order, then $d_1 \asymp g_1 \asymp (N/n)$ and $d_0 \sim g_0 \sim 1$. We also observe that $b = c + O(a\epsilon)$ which makes the problem seem very close to Arias-Castro & Verzelen (2014); Bogerd et al. (2021), although in fact the problems are quite different.

**Extension**. An extension of our alternative is that, for the $K$ communities, the sizes of $m$ of them are at the order of $N$, for an $N \ll n$ and an integer $m$, $1 \le m < K$, and the sizes of remaining $(K - m)$ are at the order of $n$. In this case, $m$ entries of $d$ are $O(N/n)$ and other entries are $O(1)$; same for $g$.

## 2.2 THE SGNQ TEST: LIMITING NULL, P-VALUE, AND POWER

In the null case, $K = 1$ and we assume $\Omega = \alpha \theta \theta'$, where $\|\theta\|_1 = n$. As $n \to \infty$, both $(\alpha, \theta)$ may vary with $n$. Write $\theta_{\max} = \|\theta\|_\infty$. We assume

$$n\alpha \to \infty, \qquad \text{and} \qquad \alpha \theta_{\max}^2 \log(n^2 \alpha) \to 0. \qquad (2.5)$$

The following theorem is adapted from Jin et al. (2021b) and the proof is omitted.

**Theorem 2.1** (Limiting null of the SgnQ statistic). *Suppose the null hypothesis is true and the regularity conditions (2.1) and (2.5) hold. As $n \to \infty$, $\psi_n \to N(0,1)$ in law.*

We have two comments. First, since the DCBM has many parameters (even in the null case), it is not an easy task to find a test statistic with a limiting null that is completely parameter free. For example, if we use the largest eigenvalue of $A$ as the test statistic, it is unclear how to normalize it so to have such a limiting null. Second, since the limiting null is completely explicit, we can approximate the (one-sided) p-value of $\psi_n$ by $\mathbb{P}(N(0,1) \ge \psi_n)$. The p-values are useful in practice, as we show in our numerical experiments.. For example, using a recent data set on the statisticians' publication (Ji et al., 2022), for each author, we can construct an ego network and apply the SgnQ test. We can then use the $p$-value to measure the co-authorship diversity of the author. Also, in many hierarchical community detection algorithms (which are presumably recursive, aiming to estimate the tree structure of communities), we can use the p-values to determine whether we should further divide a sub-community in each stage of the algorithm (e.g. Ji et al. (2022)).

The power of the SgnQ test hinges on the matrix $\widetilde{\Omega} = \Omega - (\mathbf{1}_n' \Omega \mathbf{1}_n)^{-1} \Omega \mathbf{1}_n \mathbf{1}_n' \Omega$. By basic algebra,

$$\widetilde{\Omega} = \Theta \Pi \widetilde{P} \Pi' \Theta, \qquad \text{where} \quad \widetilde{P} = P - (d' P d)^{-1} P d d' P. \qquad (2.6)$$

Let $\tilde{\lambda}_1$ be the largest (in magnitude) eigenvalue of $\widetilde{\Omega}$. Lemma 2.2 is proved in the supplement.

**Lemma 2.2.** *The rank and trace of the matrix $\widetilde{\Omega}$ are $(K-1)$ and $\|\theta\|^2 \operatorname{diag}(\tilde{P})' g$, respectively. When $K = 2$, $\tilde{\lambda}_1 = \operatorname{trace}(\widetilde{\Omega}) = \|\theta\|^2 (ac - b^2)(d_0^2 g_1 + d_1^2 g_0)/(ad_1^2 + 2bd_0 d_1 + cd_0^2)$.*

As a result of this lemma, we observe that in the SBM case, $d = h$ and thus $\tilde{\lambda}_1 = \lambda_2 \asymp N(a - c)$. To see intuitively that the power of the SgnQ test hinges on $\tilde{\lambda}_1^4/\lambda_1^2$, if we heuristically replace the terms of SgnQ by population counterparts, we obtain

$$Q_n = \sum_{i_1, i_2, i_3, i_4 (distinct)} \hat{A}_{i_1 i_2} \hat{A}_{i_2 i_3} \hat{A}_{i_3 i_4} \hat{A}_{i_4 i_1} \approx \operatorname{trace}([\Omega - \eta\eta']^4) = \operatorname{trace}(\widetilde{\Omega}^4) = \tilde{\lambda}_1^4.$$

We now formally discuss the power of the SgnQ test. We focus on the alternative hypothesis in Section 2.1. Let $d = (d_1, d_0)'$ and $g = (g_1, g_0)'$ be as in (2.2), and let $\theta_{\max,0} = \max_{i \in \mathcal{C}_0} \theta_i$ and $\theta_{\max,1} = \max_{i \in \mathcal{C}_1} \theta_i$. Suppose

$$d_1 \asymp g_1 \asymp N/n, \qquad a\theta_{\max,1}^2 = O(1), \qquad cn \to \infty, \qquad c\theta_{\max,0}^2 \log(n^2 c) \to 0. \qquad (2.7)$$

These conditions are mild. For example, when $\theta_i$'s are at the same order, the first inequality in (2.7) automatically holds, and the other inequalities in (2.7) hold if $a \le C$ for an absolute constant $C > 0$, $cn \to \infty$, and $c \log(n) \to 0$.

Fixing $0 < \kappa < 1$, let $z_\kappa > 0$ be the value such that $\mathbb{P}(N(0,1) \ge z_\kappa) = \kappa$. The level-$\kappa$ SgnQ test rejects the null if and only if $\psi_n \ge z_\kappa$, where $\psi_n$ is as in (1.3). Theorem 2.2 and Corollary 2.1 are proved in the supplement. Recall that our alternative hypothesis is defined in Section 2.1. By *power* we mean the probability that the alternative hypothesis is rejected, minimized over all possible alternative DCBMs satisfying our regularity conditions.

**Theorem 2.2** (Power of the SgnQ test). *Suppose that (2.7) holds, and let $\kappa \in (0, 1)$. Under the alternative hypothesis, if $|\tilde{\lambda}_1|/\sqrt{\lambda_1} \to \infty$, the power of the level-$\kappa$ SgnQ test tends to 1.*

**Corollary 2.1.** *Suppose the same conditions of Theorem 2.2 hold, and additionally $\theta_{\max} \le C\theta_{\min}$ so all $\theta_i$ are at the same order. In this case, $\lambda_1 \asymp cn$ and $|\tilde{\lambda}_1| \asymp N(a - c)$, and the power of the level-$\kappa$ SgnQ test tends to 1 if $N(a - c)/\sqrt{cn} \to \infty$.*

In Theorem 2.2 and Corollary 2.1, if $\kappa = \kappa_n$ and $\kappa_n \to 0$ slowly enough, then the results continues to hold, and the sum of Type I and Type II errors of the SgnQ test at level-$\kappa_n \to 0$.

The power of the SgnQ test was only studied in the balanced case (Jin et al., 2021b), but our setting is a severely unbalanced case, where the community sizes are at different orders as well as the entries of $d$ and $g$. In the balanced case, the signal-to-noise ratio of SgnQ is governed by $|\lambda_2|/\sqrt{\lambda_1}$, but in our setting, the signal-to-noise ratio is governed by $|\tilde{\lambda}_1|/\sqrt{\lambda_1}$. The proof is also subtly different. Since the entries of $P$ are at different orders, many terms deemed negligible in the power analysis of the balanced case may become non-negligible in the unbalanced case and require careful analysis.

## 2.3 COMPARISON WITH THE NAIVE DEGREE-BASED $\chi^2$-TEST

Consider a setting where $\Omega = \alpha\Theta \mathbf{1}_n \mathbf{1}_n' \Theta \equiv \alpha\theta\theta'$ under the null and $\Omega = \Theta\Pi P\Pi'\Theta$ under the alternative, and (2.1) holds. When $\theta$ is unknown, it is unclear how to apply the $\chi^2$-test: the null case has $n$ unknown parameters $\theta_1, \ldots, \theta_n$, and we need to use the degrees to estimate $\theta_i$ first. As a result, the resultant $\chi^2$-statistic may be trivially 0. Therefore, we consider a simpler SBM case where $\theta = \mathbf{1}_n$. In this case, $\Omega = \alpha \mathbf{1}_n \mathbf{1}_n'$, and $\Omega = \Pi P\Pi'$ and the null case only has one unknown parameter $\alpha$. Let $y_i$ be the degree of node $i$, and let $\hat{\alpha} = [n(n-1)]^{-1} \mathbf{1}_n' A \mathbf{1}_n$. The $\chi^2$-statistic is

$$X_n = \sum_{i=1}^{n} (y_i - n\hat{\alpha})^2/[(n-1)\hat{\alpha}(1 - \hat{\alpha})]. \qquad (2.8)$$

It is seen that as $n\alpha \to \infty$ and $\alpha \to 0$, $(X_n - n)/\sqrt{2n} \to N(0,1)$ in law. For a fixed level $\kappa \in (0, 1)$, consider the $\chi^2$-test that rejects the null if and only if $(X_n - n)/\sqrt{2n} > z_\kappa$. Let $\alpha_0 = n^{-2}(\mathbf{1}_n' \Omega \mathbf{1}_n)$. The power of the $\chi^2$-test hinges on the quantity $(n\alpha_0)^{-1}\|(\Omega\mathbf{1}_n - n\alpha_0)\|^2 = (n\alpha_0)^{-1}\|\Pi Ph - (h'Ph)^{-1}\mathbf{1}_n\|^2 = 0$, if $Ph \propto \mathbf{1}_K$. The next theorem is proved in the supplement.

**Theorem 2.3.** *Suppose $\theta = \mathbf{1}_n$ and (2.7) holds. If $|\tilde{\lambda}_1|/\sqrt{\lambda_1} \to \infty$ under the alternative hypothesis, the power of the level-$\kappa$ SgnQ test goes to 1, while the power of the level-$\kappa$ $\chi^2$-test goes to $\kappa$.*

## 2.4 THE STATISTICAL LOWER BOUND AND THE OPTIMALITY OF THE SCAN TEST

For lower bounds, it is standard to consider a random-membership DCBM (Jin et al., 2021b), where $\|\theta\|_1 = n$, $P$ is as in (2.3)-(2.4) and for a number $N \ll n$, $\Pi = [\pi_1, \pi_2, \ldots, \pi_n]'$ satisfies

$$\pi_i = (X_i, 1 - X_i), \qquad \text{where } X_i \text{ are iid Bernoulli}(\varepsilon) \text{ with } \varepsilon = N/n. \tag{2.9}$$

**Theorem 2.4** (Statistical lower bound). *Consider the null and alternative hypotheses of Section 2.1, and assume that (2.9) is satisfied, $\theta_{\max} \leq C\theta_{\min}$ and $Nc/\log n \to \infty$. If $\sqrt{N}(a-c)/\sqrt{c} \to 0$, then for any test, the sum of the type-I and type-II errors tends to $1$.*

To show the tightness of this lower bound, we introduce the signed scan test, by adapting the idea in Arias-Castro & Verzelen (2014) from the SBM case to the DCBM case. Unlike the SgnQ test and the $\chi^2$-test, signed scan test is not a polynomial time test, but it provides sharper upper bounds. Let $\hat{\eta}$ be the same as in (1.3). For any subset $S \subset \{1, 2, \ldots, n\}$, let $\mathbf{1}_S \in \mathbb{R}^n$ be the vector whose $i$th coordinate is $1\{i \in S\}$. Define the signed scan statistic

$$\phi_{sc} = \max_{S \subset \{1,2,\ldots,n\}:|S|=N} \mathbf{1}_S'\left(A - \hat{\eta}\hat{\eta}'\right)\mathbf{1}_S. \tag{2.10}$$

**Theorem 2.5** (Tightness of the statistical lower bound). *Consider the signed scan test (2.10) that rejects the null hypothesis if $\phi_{sc} > t_n$. Under the assumptions of Theorem 2.4, if $\sqrt{N}(a - c)/\sqrt{c \log(n)} \to \infty$, then there exists a sequence $t_n$ such that the sum of type I and type II errors of the signed scan test tends to $0$.*

By Theorems 2.4-2.5 and Corollary 2.1, the two hypotheses are asymptotically indistinguishable if $\sqrt{N}(a-c)/\sqrt{c} \to 0$, and are asymptotically distinguishable by the SgnQ test if $N(a-c)/\sqrt{cn} \to \infty$. Therefore, the lower bound is sharp, up to log-factors, and the signed scan test is nearly optimal. Unfortunately, the signed scan test is not polynomial-time computable. Does there exist a polynomial-time computable test that is optimal? We address this in the next section.

## 2.5 THE COMPUTATIONAL LOWER BOUND

Consider the same hypothesis pair as in Section 2.4, where $K = 2$, $P$ is as in (2.3)-(2.4), and $\Pi$ is as in (2.9). For simplicity, we only consider SBM, i.e., $\theta_i \equiv 1$. The low-degree polynomials argument emerges recently as a major tool to predicting the average-case computational barriers in a wide range of high-dimensional problems (Hopkins & Steurer, 2017; Hopkins et al., 2017). Many powerful methods, such as spectral algorithms and approximate message passing, can be formulated as functions of the input data, where the functions are polynomials with degree at most logarithm of the problem dimension. In comparison to many other schemes of developing computational lower barriers, the low-degree polynomial method yields the same threshold for various average-case hardness problems, such as community detection in the SBM (Hopkins & Steurer, 2017) and (hyper)-planted clique detection (Hopkins, 2018; Luo & Zhang, 2022). The foundation of the low-degree polynomial argument is the following *low-degree polynomial conjecture* (Hopkins et al., 2017) :

**Conjecture 2.1** (Adapted from Kunisky et al. (2019)). *Let $\mathbb{P}_n$ and $\mathbb{Q}_n$ denote a sequence of probability measures with sample space $\mathbb{R}^{n^k}$ where $k = O(1)$. Suppose that every polynomial $f$ of degree $O(\log n)$ with $\mathbb{E}_{\mathbb{Q}_n} f^2 = 1$ is bounded under $\mathbb{P}_n$ with high probability as $n \to \infty$ and that some further regularity conditions hold. Then there is no polynomial-time test distinguishing $\mathbb{P}_n$ from $\mathbb{Q}_n$ with type I and type II error tending to $0$ as $n \to \infty$.*

We refer to Hopkins (2018) for a precise statement of this conjecture's required regularity conditions. The low-degree polynomial computational lower bound for our testing problem is as follows.

**Theorem 2.6** (Computational lower bound). *Consider the null and alternative hypotheses in Section 2.1, and assume $\theta_i \equiv 1$ and (2.9) holds. As $n \to \infty$, assume $c < a$, $c < 1 - \delta$ for constant $\delta > 0$, $N < n/3$, $D = O(\log n)$, and $\limsup_{n \to \infty} \left\{ \left( \log_n \frac{N}{\sqrt{n}} + \log_n \frac{a-c}{\sqrt{c}} \right) \vee \left( \sqrt{D/2 - 1} \log_n \frac{a-c}{\sqrt{c}} \right) \right\} < 0$. For any series of degree-$D$ polynomials $\phi_n : A \to \mathbb{R}$, whenever $\mathbb{E}_{H_0} \phi_n(A) = 0$, $Var_{H_0}(\phi_n(A)) = 1$, we must have $\mathbb{E}_{H_1} \phi_n(A) = o(1)$. This implies if Conjecture 2.1 is true, there is no consistent polynomial-time test for this problem.*

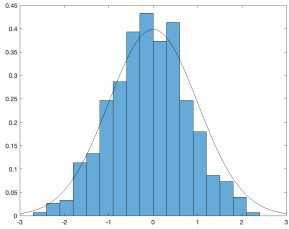 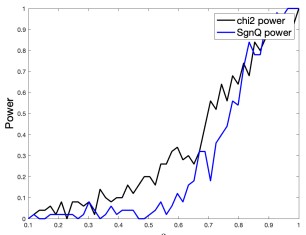 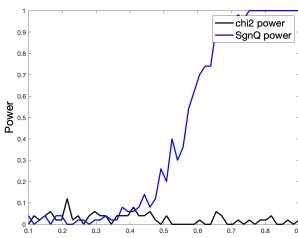

Figure 2: Left: Null distribution of SgnQ ($n = 500$). Middle and right: Power comparison of SgnQ and $\chi^2$ ($n = 100$, $N = 10$, 50 repetitions). We consider a 2-community SBM with $P_{11} = a$, $P_{22} = 0.1$, $P_{12} = 0.1$ (middle plot) and $P_{12} = \frac{an-(a+0.1)N}{n}$ (right plot, the case of degree matching).

By Theorem 2.6, if both $(a - c)/\sqrt{c} \lesssim 1$ and $N(a - c)/\sqrt{cn} \to 0$, the testing problem is computationally infeasible. The region where the testing problem is statistically possible but the SgnQ test loses power corresponds to $N(a - c)/\sqrt{cn} \to 0$. If $N \gtrsim \sqrt{n}$, Theorem 2.6 already implies that this is the computationally infeasible region; in other words, SgnQ achieves the CLB and is optimal. If $N = o(\sqrt{n})$, SgnQ solves the detection problem only when $(a - c)/\sqrt{c} \gg n^{1/2}$, i.e. when the node-wise SNR is strong. We discuss the case of moderate node-wise SNR in the next subsection.

## 2.6 THE POWER OF EST, AND DISCUSSIONS OF THE TIGHTNESS OF CLB

When $N = o(\sqrt{n})$ and $(a - c)/\sqrt{c} \to \infty$ both hold, the upper bound by SgnQ does not match with the CLB. It is unclear whether the CLB is tight. To investigate the CLB in this regime, we consider other possible polynomial-time tests. The economic scan test (EST) is one candidate. Given fixed positive integers $v$ and $e$, the EST statistic is defined to be $\phi_{EST}^{(v)} \equiv \sup_{|S| \leq v} \sum_{i,j \in S} A_{ij}$, and the EST is defined to reject if and only if $\phi_{EST}^{(v)} \geq e$. EST can be computed in time $O(n^v)$, which is polynomial time. For simplicity, we consider the SBM, i.e. where $\theta = \mathbf{1}_n$, and a specific setting of parameters for the null and alternative hypotheses.

**Theorem 2.7** (Power of EST). *Suppose $\beta \in [1/2, 1)$ and $0 < \omega < \delta < 1$ are fixed constants. Under the alternative, suppose $\theta = \mathbf{1}_n$, (2.9) holds, $N = n^{1-\beta}$, $a = n^{-\omega}$, and $c = n^{-\delta}$. Under the null, suppose $\theta = \mathbf{1}_n$ and $\alpha = a(N/n) + b(1 - N/n)$. If $\omega/(1 - \beta) < \delta$, the sum of type I and type II errors of the EST with $v$ and $e$ satisfying $\omega/(1 - \beta) < v/e < \delta$ tends to 0.*

Theorem 2.7 follows from standard results in probabilistic combinatorics (Alon & Spencer, 2016). It is conjectured in Bhaskara et al. (2010) that EST attains the CLB in the Erdös-Renyi setting considered by Arias-Castro & Verzelen (2014); Verzelen & Arias-Castro (2015). This suggests that the CLB in Theorem 2.6 is likely not tight when $N = o(\sqrt{n})$ and $(a - c)/\sqrt{c} \to \infty$. However, this is not because our inequalities in proving the CLB are loose. A possible reason is that the prediction from the low-degree polynomial conjecture does not provide a tight bound. It remains an open question whether other computational infeasibility frameworks provide a tight CLB in our problem.

## 2.7 THE PHASE TRANSITION

We describe more precisely our results in terms of the phase transitions shown in Figure 1. Consider the null and alternative hypotheses from Section 2.1. For illustration purposes, we fix constants $\beta \in (0, 1)$ and $\gamma \in \mathbb{R}$ and assume that $N = n^{1-\beta}$ and $(a - c)/\sqrt{c} = n^{-\gamma}$. In the two-dimensional space of $(\gamma, \beta)$, the region of $\beta > 1/2$ and $\beta < 1/2$ corresponds to that the size of the small community is $\gg \sqrt{n}$ and $o(\sqrt{n})$, respectively, and the regions of $\gamma > 0$, $-1/2 < \gamma < 0$ and $\gamma < -1/2$ correspond to 'weak node-wise signal', 'moderate node-wise signal,' and the 'strong node-wise signal', respectively. See Figure 1. By our results in Section 2.4, the testing problem is statistically impossible if $\beta + 2\gamma > 1$ (orange region). By our results in Section 2.2, SgnQ has a full power if $\beta + \gamma < 1/2$ (blue region). Our results in Section 2.5 state that the testing problem is computationally infeasible if both $\gamma > 0$ and $\beta + \gamma > 1/2$ (green and orange regions). Combining these results, when $\beta < 1/2$, we have a complete understanding of the LB and CLB.

## 3 NUMERICAL RESULTS

**Simulations**. First in Figure 2 (left panel) we demonstrate the asymptotic normality of SgnQ under a null of the form $\Omega = \theta\theta'$, where $\theta_i$ are i.i.d. generated from $\mathrm{Pareto}(4, 0.375)$. Though the degree heterogeneity is severe, SgnQ properly standardized is approximately standard normal under the null. Next in Figure 2 we compare the power of SgnQ in an asymmetric and symmetric SBM model. As our theory predicts, both tests are powerful when degrees are not calibrated in each model, but only SgnQ is powerful in the symmetric case. We also compare the power of SgnQ with the scan test to show evidence of a statistical-computational gap. We relegate these experiments to the supplement.

**Real data**: Next we demonstrate the effectiveness of SgnQ in detecting small communities in coauthorship networks studied in Ji et al. (2022). In Example 1, we consider the personalized network of Raymond Carroll, whose nodes consist of his coauthors for papers in a set of 36 statistics journals from the time period 1975 – 2015. An edge is placed between two coauthors if they wrote a paper in this set of journals during the specified time period. The SgnQ p-value for Carroll's personalized network $G_{\mathsf{Carroll}}$ is 0.02, which suggests the presence of more than one community. In Ji et al. (2022), the authors identify a small cluster of coauthors from a collaboration with the National Cancer Institute. We applied the SCORE community detection module with $K = 2$ (e.g. Ke & Jin (2022)) and obtained a larger community $G_{\mathsf{Carroll}}^0$ of size 218 and a smaller community $G_{\mathsf{Carroll}}^1$ of size 17. Precisely, we removed Carroll from his network, applied SCORE on the remaining giant component, and defined $G_{\mathsf{Carroll}}^0$ to be the complement of the smaller community. The SgnQ p-values in the table below suggest that both $G_{\mathsf{Carroll}}^0$ and $G_{\mathsf{Carroll}}^1$ are tightly clustered. Refer to the supplement for a visualization of Carroll's network and its smaller community labeled by author names. In Example 2, we consider three different coauthorship networks $G_{\mathsf{old}}$, $G_{\mathsf{recent}}$, and $G_{\mathsf{new}}$ corresponding to time periods (i) 1975-1997, (ii) 1995-2007, and (iii) 2005-2015 for the journals AoS, Bka, JASA, and JRSSB. Nodes are given by authors, and an edge is placed between two authors if they coauthored at least one paper in one of these journals during the corresponding time period. For each network, we perform a similar procedure as in the first example. First we compute the SgnQ p-value, which turns out to be $\approx 0$ (up to 16 digits of precision) for all networks. For each $i \in \{\mathsf{old}, \mathsf{recent}, \mathsf{new}\}$, we apply SCORE with $K = 2$ to $G_i$ and compute the SgnQ p-value on both resulting communities, let us call them $G_i^0$ and $G_i^1$. We refer to the table below for the results. For $G_{\mathsf{old}}$ and $G_{\mathsf{recent}}$, SCORE with $K = 2$ extracts a small community. The SgnQ p-value further supports the hypothesis that this small community is well-connected. In the last network, SCORE splits $G_{\mathsf{new}}$ into two similarly sized pieces whose p-values suggests they can be split into smaller subcommunities.

| Example | Network | Size | SgnQ p-value | Communities | Sizes | SgnQ p-values |
|---------|---------|------|--------------|-------------|-------|---------------|
| 1 | $G_{\mathsf{Carroll}}$ | 235 | 0.02 | $(G_{\mathsf{Carroll}}^0, G_{\mathsf{Carroll}}^1)$ | (218, 17) | (0.134, 0.682) |
| 2 | $G_{\mathsf{old}}$ | 2647 | 0 | $(G_{\mathsf{old}}^0, G_{\mathsf{old}}^1)$ | (2586, 61) | (0, 0.700) |
| | $G_{\mathsf{recent}}$ | 2554 | 0 | $(G_{\mathsf{recent}}^0, G_{\mathsf{recent}}^1)$ | (2540,14) | (0, 0.759) |
| | $G_{\mathsf{new}}$ | 2920 | 0 | $(G_{\mathsf{new}}^0, G_{\mathsf{new}}^1)$ | (1685,1235) | (0, 0) |

**Discussions**: Global testing is a fundamental problem and often the starting point of a long line of research. For example, in the literature of Gaussian models, certain methods started as a global testing tool, but later grew into tools for variable selection, classification, and clustering and motivated many researches (e.g., Donoho & Jin (2004; 2015)). The SgnQ test may also motivate tools for many other problems, such as estimating the locations of the clique and clustering more generally. For example, in Jin et al. (2022), the SgnQ test motivated a tool for estimating the number of communities (see also Ma et al. (2021)). SgnQ is also extendable to clique detection in a tensor (Yuan et al., 2021; Jin et al., 2021a) and for network change point detection. The LB and CLB we obtain in this paper are also useful for studying other problems, such as clique estimation. If you cannot tell whether there is a clique in the network, then it is impossible to estimate the clique. Therefore, the LB and CLB are also valid for the clique estimation problem (Alon et al., 1998; Ron & Feige, 2010).

The limiting distribution of SgnQ is $N(0, 1)$. This is not easy to achieve if we use other testing ideas, such as the leading eigenvalues of the adjacency matrix: the limiting distribution depends on many unknown parameters and it is hard to normalize (Liu et al., 2019). The p-value of the SgnQ test is easy to approximate and also useful in applications. For example, we can use it to measure the research diversity of a given author. Consider the ego sub-network of an author in a large co-authorship or citation network. A smaller p-value suggests that the ego network has more than 1 communitiy and has more diverse interests. The p-values can also be useful as a stopping criterion in hierarchical community detection modules.

**Acknowledgments.** We thank the anonymous referees for their helpful comments. We thank Louis Cammarata for assistance with the simulations in Section A.3. J. Jin was partially supported by NSF grant DMS-2015469. Z.T. Ke was supported in part by NSF CAREER Grant DMS-1943902. A.R. Zhang acknowledges the grant NSF CAREER-2203741.

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
