# OpenReview forum: "Phase transition for detecting a small community in a large network"
_ICLR.cc/2023/Conference — ICLR 2023 poster_

### Official Review · Reviewer_US3K · 2022-10-17

**Confidence:** 4
**Correctness:** 4
**Technical Novelty And Significance:** 3
**Empirical Novelty And Significance:** Not applicable
**Recommendation:** 6

**Clarity, Quality, Novelty And Reproducibility:**

See above for quality/novelty of results. The difference of power between the $\chi^2$ and SgnQ tests are nicely illustrated, with figures showing the distinction between a symmetric and non-symmetric SBM.

Here is the more precise feedback on the paper clarity:

**Presentation**:
- the relevant model definitions are split between the very beginning of the introduction and section 2.1, which makes it hard to refer to. In general, the introduction is a bit rushed; the Sinkhorn-type argument is introduced before the $\chi^2$ test is properly defined. I think the paper would benefit from a more formal model introduction, which defines all the useful quantities used in the theorems.
- the paper is overall very technical, and provides almost no rationale behind all the definitions and algorithms. Some questions/remarks:
  - $\tilde \Omega, \tilde P, \hat A$ (among others) are defined in matrix form, and it is hard to understand what they correspond to (you basically zero out the image of a specific vector, but why this one?). More minorly, for consistency $\hat A$ should maybe be renamed to $\tilde A$.
  - accordingly, the intuition behind the SgnQ algorithm based on $\hat A$ is never given.
  - the vector $h$ is defined twice, in slightly different ways: in the introduction, it is given as $\mathbb E[\pi]$, while in section 2.1 it is the empirical vector $\Pi \mathbf 1$ (this matrix characterization should probably also be given, since it is useful to understand some of the results afterwards)
  - I tend to agree that $d$ and $g$ are quite natural vectors, but their significance should be expanded upon; for example, the fact that $d = h$ in the SBM case is quite crucial, since it means that $\tilde \lambda_1 = \lambda_2$.
  - the notion of "power" of the tests should be explicitly defined.
  - for (2.3) and after, why are the community labels swapped (i.e. why does $\mathcal C_1$ correspond to the first community and $\mathcal C_0$ to the second ?).
  - Lemma 2.2 does not give much insight into the problem, and should probably be relegated in the appendix. Alternatively, it would be more meaningful to expand on why this means that $\tilde \lambda_1 \simeq N(a-c)$
  - when writing $Ph \propto \mathbf 1_K$, I think it would be interesting to make the proportionality constant explicit (for notation consistency, let's call it $\alpha$) and mention that $n\alpha$ is the average degree of the graph, just like in the null model. This would give people used to the SBM a better grasp of the degree conditions such as the ones in (2.5) and (2.7).
  - the notation $'$ for matrix transpose is not very standard, so it should be made explicit somewhere.

**Proof**: the proofs of appendix E rely too much on Jin et al. for key quantities. Of course, lifting bounds is perfectly normal, but the quantities of interest ($X_i, Y_i, Z_i$) should be defined in a self-contained way. Even more important, no definition for the *ideal*, *proxy* and *real* versions of $Q$ are given, so the structure of the proof cannot be followed. Otherwise, the proof is very computational, so I didn't every inch of it; but Lemma E.13 is a welcome formalism.



**Minor remarks/typos**:

- in (2.4), I found interesting that $b = c + O(\epsilon)$, this may be worth to mention, in relation with the classical planted clique setting
- above (2.6), the matrix $G$ is never used
- I assumed that $(dist)$ means that the summation terms need to be distinct; if so this should be made explicit
- in E.4.1, the definition of T should have $i_m$ and not $i_M$; is it the same $m$ as in the definition of the DCBM model ?
- below Theorem F.1, the conditions (i)-(iv) should be (1.)-(4.)
- the checking of conditions from Appendix C and Lemma E1 can probably be merged, since it is the same change of parametrization
- the use of $\Omega$ in appendix I is very confusing

**Strength And Weaknesses:**

The setting considered by the paper is both natural and interesting; it is closer to a real-world setting than the vanilla planted clique problem, and therefore the failure of $\chi^2$ and the need for other methods are meaningful. The authors obtain a near complete phase diagram, with only a small gap in the $N \gg \sqrt{n}$ regime.

However, the main weakness of the paper is the overall clarity and presentation of the results; precise feedback is given in the next section. In particular, it is nearly impossible to check the proof without constantly referring to Jin et al. for every single definition.

It is also my opinion that this paper would be more suited for a more theoretical CS conference/paper, although I do not consider it grounds for rejection.

**Summary Of The Paper:**

This paper studies the task of detecting a planted community structure with small clusters (e.g. a planted clique) inside a graph with heterogeneous degrees. More precisely, the authors consider a random graph model such that
$$ P(i \sim j) = \theta_i \theta_j P_{\pi(i), \pi(j)}  $$
where $\theta$ models the degree heterogeneity between vertices, and $\pi$ is an assignment of vertices into $K$ different communities. The goal is to find a test that discriminates between the null hypothesis $K = 1$ and its alternative, even in the presence of very small communities. In particular, the alternative model considered consists in taking $K=2$, and two communities of size $n$ and $N \ll n$, with the matrix $P$ given as
$$ P = \\begin{bmatrix} a & b
\\\\ b & c \\end{bmatrix} $$

The paper's contributions are threefold:
- **Existence of a polynomial test:** consider the *signed quadrilateral* statistic
$$ \psi_n = \sum_{i_1, \dots, i_4} \hat A_{i_1 i_2} \hat A_{i_2 i_3} \hat A_{i_3 i_4} \hat A_{i_4 i_1},  $$
where $\hat A$ is a centered version of $A$. The authors show that there exists a certain threshold $\tau$ depending on the eigenvalues of $\mathbb E[A]$ (and related matrices), such that if $\tau\to\infty$ there exists a test based on the value of $\psi_n$ that discriminated between $K = 1$ and $K > 1$ whp. In the two-community case above, this corresponds to
$$ \tau = N(a-c)/\sqrt{cn}. $$

- **Statistical exact threshold:** in the two-community case, this paper show the existence of a statistical threshold at
$$\tau' = \sqrt{N}(a-c)/\sqrt{c} $$ such that if $\tau' \to 0$, no test can distinguish between $K=1$ and $K>1$. Conversely, if $\tau' \to \infty$, then there exists a (non-polynomial) test that discriminates between the two hypotheses.

- **Computational lower bound:** finally, if $\tau \to 0$, the authors provide a computational lower-bound based on the low-degree framework, showing that there is no succesfull test based on a low-degree polynomial in $A$. This is a proxy for polynomial hardness. Since $\tau$ and $\tau'$ differ significantly only when $N \ll \sqrt{n}$, this implies that the signed quadrilateral test is optimal above $\sqrt{n}$.

**Summary Of The Review:**

This paper presents very interesting results on testing a generalized version of the planted clique setting, which is closer to real-world applications. However, the presentation must be significantly improved, and the proof made self-contained, hence I do not recommend acceptance.

---

> ### Author Response · Authors · 2022-11-19
> **Response to Reviewer US3K**
>
> Thank you for providing such a thorough summary of our findings. We are glad that you find our setting is "both natural and interesting" and that you recognize our paper's solid theoretical contributions.
>
> We are surprised to see a discrepancy between your score and your positive assessment of our contributions. You expressed two concerns, one about the paper's presentation and the other about making proofs self-contained. We think both can be addressed (please see our point-to-point response below). Since you agreed that our paper provides solid theoretical results for an interesting problem, we hope to be given the opportunity to revise and publish it.
>
> ---
> **Intuition for SgnQ**:
>
>  In works prior to SgnQ, polygon-based statistics had been introduced and shown to be effective in some network testing problems; for example in [Bubeck et al]. The polygon statistics are natural higher-moment generalizations of total-edge-based tests and degree-based-tests that appear in [Arias-Castro & Verzelen], for example. A key discovery of [Jin, Ke, Luo] is that a proper centering of the network, by subtracting away a naive degree estimate (namely $\hat \eta \hat \eta'$), leads to much better performance of polygon-based statistics in sparse settings where the degrees grow very slowly. This leads to the signed polygon statistic studied in our work. Perhaps surprisingly, [3] shows that SgnQ performs optimally in the balanced setting even when the degree estimate $\hat \eta$ is very crude, which is certainly the case in the degree-corrected block model.
>
> To gain further intuition for the performance of SgnQ, we note the following:
>
> - Under the null, it is known that $\mathrm{Var}(Q_n) \approx (\|\widehat{\eta}\|^2 - 1)^4 \approx \lambda_1^4$ (see [3])
> - Next, if we (heuristically) replace the terms of SgnQ with their population counterparts, we obtain
>  	\begin{align*}
>  		Q_n &= \sum_{\substack{i_1, i_2, i_3, i_4 (distinct)}}  (A_{i_1 i_2} - \hat \eta_{i_1} \hat \eta_{i_2}  ) (A_{i_2 i_3} - \hat \eta_{i_2} \hat \eta_{i_3}  )
>  		(A_{i_3 i_4} - \hat \eta_{i_3} \hat \eta_{i_4}  )
>  		(A_{i_4 i_1} - \hat \eta_{i_4} \hat \eta_{i_1}  )
>  		\\ &
>  		\approx \mathrm{trace}([\Omega - \eta \eta']^4)
>  		= \mathrm{trace}(\widetilde \Omega^4)
>  		= \tilde \lambda_1^4
>  	\end{align*}
> - Therefore, the power of the SgnQ test hinges on $
>  	$\tilde \lambda_1^4/\lambda_1^2$.
>
> In the balanced case, it is known that $\tilde \lambda_1 \asymp \lambda_2(\Omega)$. Thus, SgnQ captures the signal from key spectral information that is tied directly to the community structure.
>
>
> ---
>
> **Proofs**:
>
> Again, thanks very much for the detailed suggestions here and the careful read. We have incorporated your suggested changes into our revision. In particular, every term is defined explicitly (including ideal, proxy, and real SgnQ along with their decompositions into post-expansion sums), so our proof is now entirely self-contained. We're glad that you also feel that it's reasonable to borrow bounds and identities from [3] when possible, and our argument still uses this quite a bit. Nevertheless, we are very careful to be explicit  down to the page number when borrowing results from [3].
>
> ---
>
> **Writing suggestions**:
>
> - Organization: We have done our best to keep the first section light in the revision and then follow up with precise model definitions and results in Section 2.1.
>
> - Regarding $\tilde \Omega, \tilde P,...$: We zero out this specific vector in $\tilde \Omega$ because it is the population counterpart of the one that is subtracted off in the definition of $\hat A$ that is used to form SgnQ. We have added some comments about this in Section 1.1.
>
> - Regarding $\mathcal{C}_0, \mathcal{C}_1$: We chose to follow the convention in Arias-Castro--Verzelen for consistency.
>
> - Regarding transpose: We added a footnote on page 2.
>
> - We kindly thank you for the remaining suggestions and will incorporate them in a later revision.
>
> ---
>
> **Minor comments**:
>
> -  $b = c + O(\epsilon)$: Thanks for pointing this out, we added a comment below (2.4).
>
> - Regarding (dist): We've added a clarification above (1.3).
>
> - The definition of $T$: This was a typo and it should be $m$ instead of $M$. The $m$ here is not related to the one in the DCBM definition. Since we are only studying SgnQ, we may just take $m = 4$, though the framework of Lemma E.13 would apply to higher-order signed polygon tests as well, so we state it in a general way.
>
>  - Theorem F.1: We fixed it to (i)--(iv)
>
>  - Use of $\Omega$ in Appendix I: We have replaced $\Omega$ with $\Gamma$.
>
>   - We kindly thank you for the remaining suggestions and will incorporate them in a later revision.

---

> > ### Comment · Reviewer_US3K · 2022-11-28
> > **Response to the authors**
> >
> > Thank you for all the revisions brought to the manuscript, in particular the proof cleanup. I have raised my score accordingly to recommend acceptance. A further revision might be needed to add some improvements to the presentation still.

---

> > > ### Author Response · Authors · 2022-12-01
> > > **Thank you for the feedback**
> > >
> > > Thank you for your reply. We are delighted that you raised the score. Also, thank you for your excellent suggestions for improving writing. If you have other comments or suggestions that you wish to share, please feel free to let us know. We are pleased to incorporate them in a further revision.

---

### Official Review · Reviewer_JaWW · 2022-10-24

**Confidence:** 3
**Correctness:** 4
**Technical Novelty And Significance:** 3
**Empirical Novelty And Significance:** 3
**Recommendation:** 6

**Clarity, Quality, Novelty And Reproducibility:**

This paper is not very easy to follow due to the writing and arrangement. This paper analyzes this existing SgnQ test in a new small community detection problem.

**Strength And Weaknesses:**

Strengths:
1. The small community detection problem is well-motivated. They show the naive degree-based $\chi^2$ test is powerless for some scenarios.
2. They provide the phase transitions for this testing problem. The unbalanced community detection is harder than the balanced version. They show SgnQ test still works for unbalanced community detection in some regimes.

Weaknesses:
1. Does the identifiable issue only happen for the degree-corrected block model? I think it would also hold for the stochastic block model.
2. I think some explanations like general settings with more than two communities and normalization in section 2.1 are not helpful. It makes the setting too complicated to follow.
3. Although it is mentioned it is not easy to analyze the SgnQ test in the unbalanced setting, I think it would be very helpful to see a more intuitive explanation of how the analysis works.

**Summary Of The Paper:**

This paper considers community detection with unbalanced communities. Specifically, they consider the hypothesis testing problem on the degree-corrected block model, which is to distinguish whether a graph is generated from a null model with one community or an alternative model with unbalanced communities.

They show phase transitions of this testing problem. When the size of the small community is larger than $\sqrt{n}$, they show that the SgnQ test (Jin et al. 2021) is optimal. They show the matching computational lower bound in this case. When the size of the small community is smaller than $\sqrt{n}$, they show the parameter regime where the SgnQ test works.

**Summary Of The Review:**

Overall, this paper is very interesting. However, the main techniques to make the SgnQ test work for the unbalanced community detection is unclear to me since it is not well explained in the main paper.

---

> ### Author Response · Authors · 2022-11-19
> **Response to Reviewer JaWW**
>
> Thank you for the thoughtful comments and questions. We respond to them point-by-point below.
>
> ---
>
> **Identifiability**: For SBM, we can express the distribution only in terms of $a, b,$ and $c$, so we do not need the heterogeneity parameter $\theta$. In this case, there is no identifiability issue.
>
> However, this SBM can be expressed in terms of DCBM in many different ways by considering rescalings of $a, b, c$ and $\theta$. For that reason, we consider a specific parametrization of DCBM as described in Section 2.1 such that every DCBM distribution can be expressed by a unique set of parameters (thus removing the identifiability issue).
>
> ---
>
> **Analysis of SgnQ**:
>
> The power analysis of SgnQ follows by showing that its mean under the alternative is much larger than its standard deviation. We take an intuitive look at this below.
>
> - Under the null, it is known that $\mathrm{Var}(Q_n) \approx (\|\widehat{\eta}\|^2 - 1)^4 \approx \lambda_1^4$ (see [1])
>
> - Next, if we (heuristically) replace the terms of SgnQ with their population counterparts, we obtain
> 	\begin{align*}
> 		Q_n &= \sum_{\substack{i_1, i_2, i_3, i_4 (distinct)}}  (A_{i_1 i_2} - \hat \eta_{i_1} \hat \eta_{i_2}  ) (A_{i_2 i_3} - \hat \eta_{i_2} \hat \eta_{i_3}  )
> 		(A_{i_3 i_4} - \hat \eta_{i_3} \hat \eta_{i_4}  )
> 		(A_{i_4 i_1} - \hat \eta_{i_4} \hat \eta_{i_1}  )
> 		\quad
> 		\approx \mathrm{trace}([\Omega - \eta \eta']^4) = \mathrm{trace}(\widetilde \Omega^4)
> 	\geq \tilde{\lambda}_1^4
> 	\end{align*}
>
> - Therefore, the power of the SgnQ test hinges on $
> 	\tilde \lambda_1^4/\lambda_1^2$.
>
> Making this heuristic rigorous is not an easy task. [Jin, Ke, Luo] perform this mean and variance calculation in the balanced community size case by decomposing SgnQ into $4^4 = 256$ terms. The total number of terms to analyze is reduced by symmetry considerations, but still at least $100$ terms remain. Our analysis borrows their decomposition directly. However, the bounds that [Jin, Ke, Luo] use for many of the terms are suboptimal, because they do not properly leverage the structure of the small community. Our main analytic contribution is to produce tractable and tight enough bounds for all of these terms in the severely unbalanced case.
>
> ---
>
> **Writing suggestions**: We agree that the writing can be improved. Our model is quite technical, so a degree of formality is required, although in the revision we have tried our best to balance this with enough conceptual understanding and intuition.  If there are other suggestions you have to improve the writing, we would greatly appreciate them.

---

> > ### Comment · Reviewer_JaWW · 2022-12-06
> > **Response**
> >
> > Thanks for the authors' responses and other reviewers' comments.
> >
> > The authors' responses answered my question about identifiability. The authors' responses on the analysis of SgnQ also made it easier to understand the main techniques used in the paper.

---

> > > ### Author Response · Authors · 2022-12-06
> > > **Thanks for the reply**
> > >
> > > Thank you for the reply. We are glad that our response clarified your questions. We will incorporate these explanations into our next revision and also welcome any further suggestions for improvement.

---

### Official Review · Reviewer_dBYC · 2022-10-25

**Confidence:** 2
**Correctness:** 3
**Technical Novelty And Significance:** 2
**Empirical Novelty And Significance:** 2
**Recommendation:** 6

**Clarity, Quality, Novelty And Reproducibility:**

- This paper suffers from mathiness, especially in the introduction. To me, for one example, this paragraph right on page 2 "To see why the signal captured ..." is not readable. Few will have an idea of what is going on behind the wall of notations at this point. Ironically, this paragraph serves as an important motivating example for the whole paper. I would suggest the authors to consider rephrase that part in their own words instead of throwing math notations.

- The organization of the paper is another issue. Is it really necessary to introduce every definition, model, example, statistic in the introduction? On the other hand, why not put Figure 2 (which should be the most important message in the paper) to an earlier part?

**Strength And Weaknesses:**

- I don't know how technically novel this paper is. A lot of analysis are similar to those in [Arias-Castro & Verzelen], [Ji et al.], except considering on an arguably more general model DCBM. The authors highlight in the introduction, that their problem is more challenging because they need to consider the computational gap. That is not entirely new because it is well-studied for the SBM (e.g. [1]). Could the authors highlight their technical novelties? What can other researchers learn from this paper if they want to work on similar models?

- Section 2.5: Why we only consider the case of SBM instead of more general cases here? Is it because of any difficulty in analysis?

- Figure 2: How should I interpret the white region (\beta > 0.5)? Is it because of any difficulty or limitation in analysis?

- Figure 2: I note that Theorem 2.4 (statistical lower bound, orange region) and Theorem 2.2 (SgnQ upper bound, blue region) all talk about the general DCBM case, while Theorem 2.6 (computational lower bound, green region) talks about the specific SBM case. The paper claims to establish the phase transitions for DCBM. How does that connect to the SBM assumption (the computational lower bound for SBM is already known [1])? Is that sufficient for the claim? This could be a writing issue but I don't know if there is any hidden assumption here, and if yes it should be pointed out in the figure.

- Does the real-world dataset follow the DCBM? Why DCBM is a reasonable ground assumption in this case?


References:
[1] Chen, Yudong, and Jiaming Xu. "Statistical-computational phase transitions in planted models: The high-dimensional setting." International conference on machine learning. PMLR, 2014.

**Summary Of The Paper:**

The paper studies phase transition for detecting communities under the degree-corrected block model (DCBM). The motivation is that chi-square test is not good for DCBM because of artifacts, so they use the the Signed-Quadrilateral (SgnQ) instead. The authors provide the SgnQ efficient regime, the computationally infeasible regime, as well as the statistically impossible regime for the model. Some simulations and experiments are included.

**Summary Of The Review:**

The technical novelty of this paper is unclear. The writing can be hard to follow, especially in the introduction.

---

> ### Author Response · Authors · 2022-11-19
> **Response to Reviewer dBYC (Part 1)**
>
> Thank you for your comments and for a nice summary of our results.
>
> ---
> **Why moving from SBM to DCBM makes a big difference:**
>
> It is *not* the case that we simply repeat the analysis of SBM on a more general model. In fact, moving from SBM to DCBM fundamentally changes the testing problem.
> - *The null hypothesis becomes highly composite.* Under SBM, the null model reduces to an Erdos-Renyi model, and there is only one unknown parameter. However, under DCBM, the null model reduces to $\Omega_{ij}=\theta_i\theta_j$, which still consists of a large number of unknown parameters. This means many testing strategies for SBM are not applicable here. The pool of tests we can consider is much smaller.
>
> - *The phase transition has fundamentally changed.* A common misunderstanding is that if we let $\theta_i$ be equal, the results for DCBM will always reduce to the results for SBM. This is incorrect for testing problem, because the model class we consider significantly changes the least-favorable configurations. When we consider the testing problem for a larger model class, it is easier to construct a pair of null/alternative hypotheses that are asymptotically indistinguishable (e.g., in DCBM, we can use degree matching to construct such pairs, but this can be infeasible for SBM).
>
> - The computational limits of the testing problem will also be different, as a consequence of a smaller pool of computable tests and the change of statistical limits.
>
> ---
>
> **Comment**: Technical novelty. Comparison with [Arias-Castro & Verzelen], [Jin et al.] and [Chen and Xu]
>
> **Response**: As we have explained, the testing problem changes fundamentally from SBM to DCBM. Since we use different tests and the target-to-prove lower bounds are also different,  we need technical novelties to prove the new results.
>
> -- Comparison with [Arias-Castro & Verzelen]
>
> [Arias-Castro & Verzelen] considers a 2-community SBM model. One regime of the statistical lower bound is matched by a degree-based chi-square test. However, we discover that the chi-square test is powerless under the more general DCBM. Therefore, the statistical/computational lower bounds have changed in this regime. We also need different tests to match these lower bounds. The proofs of upper/lower bounds are both very different from those of [Arias-Castro & Verzelen].
>
> -- Comparison with [Jin, Ke & Luo]
>
> As we have mentioned, there are not many tests available for DCBM. The SgnQ test in [Jin, Ke & Luo] is one of the few we can consider. However, the SgnQ test was only studied in the setting of balanced community sizes. In the severely unbalanced setting, there is no existing result about the power of SignQ and whether it matches the statistical/computational lower bounds. Our results highlight: 1) In the severely unbalanced setting, the signal-to-noise ratio of SgnQ takes a different form compared with the balanced setting. 2) In the balanced setting considered by [Jin, Ke & Luo], there is no statistical-to-computational gap, but we prove such a gap in the severely unbalanced setting.
>
> --- Comparison with [Chen and Xu] and [Hajek, Wu & Xu]
>
> [Chen and Xu] only conjectured the statistical-to-computational gap but did not prove it.
>
> We instead compare our result with [Hajek, Wu & Xu]. First, their model is much simpler than ours. Particularly, when $r=1$ in their model, it is the same as the model in [Arias-Castro & Verzelen]. As we have explained above, our statistical limits and computational limits are both different from theirs (even letting $\theta_i$ be equal in our results).  Second, the proofs of computational lower bounds are different. Their proof is based on a reduction to the planted clique conjecture. This technique is not sufficient for proving our computational lower bound. We use a much more recent technique via a reduction to the low-degree polynomial conjecture (please see our Section 2.5).

---

> ### Author Response · Authors · 2022-11-19
> **Response to Reviewer dBYC (Part 2)**
>
> (Response continues here)
>
> ---
> **Comment**: "Section 2.5: Why only consider the case of SBM instead of more general cases here? Is it because of any difficulty in analysis?"
>
> **Response**: This is not because of technical difficulty. The reason is that a computational lower bound on SBM here is already sufficient for our argument.
>
> By definition of "lower bounds", any lower bound for a smaller class of models is also a valid lower bound for the larger class of models, except that it may not be tight. Therefore, it is perfectly okay to derive the lower bound on a narrower model class and use it for a wider model class, as long as it can match the upper bound on the wider model class (i.e., being "tight"). It is the case here, where the computational lower bound derived under SBM happens to match the computational upper bound for DCBM, hence, it suffices for our purpose.
>
> (We clarify that this does not mean the computational limits of SBM and DCBM are the same. They are only the same in some regions, such as the one considered in Section 2.5.)
>
> ---
> **Comment**: "Figure 2: How should I interpret the white region (\beta > 0.5)? Is it because of any difficulty or limitation in analysis?"
>
> **Response**:  Yes, this is partly because of difficulty in analysis. In that white region of Figure 2 (which is now Figure 1 of the revised draft), the computational upper bound and the computational lower bound do not match. One or both of them need improvement. We conjecture that the computational lower bound needs improvement, because we find a economic scan test (EST) that is polynomial-time and has power in a sub-region of this white region. In our proof of computational lower bounds, we have already used the up-to-date techniques (i.e., reduction to the low-degree polynomial conjecture). A further improvement may call for more advanced techniques, which is, to the best of our knowledge, not yet available in the literature.
>
> ---
> **Comment**: "The paper claims to establish the phase transitions for DCBM. How does that connect to the SBM assumption (the computational lower bound for SBM is already known [1])? Is that sufficient for the claim?"
>
> **Response**: As we have explained, the phase transitions for DCBM are different from those of SBM. The computational lower bounds for two settings are not always the same. Also, the proof technique in [Hajek, Wu & Xu] is based on a reduction to the planted clique conjecture, which is insufficient to prove our computational lower bounds. We need to use the more recent technique of a reduction to the low-degree polynomial conjecture.
>
> ---
>
> **Comment**:  "Does the real-world dataset follow the DCBM? Why DCBM is a reasonable ground assumption in this case?"
>
> **Response**: This is a great point. There have been many evidences showing that DCBM is a reasonable model for real-world networks. For example, please see Ji, Jin, Ke and Li (2021). Co-citation and co-authorship networks of statisticians.
>
> ---
>
> **Comment**: Suggestions for improving writing.
>
> Thank you for making these great suggestions. We have taken them to improve this paper.
>
> --- ``To see why the signal captured...": We have simplified this part in the introduction and moved a revised version of the more detailed argument to Section 2.1.
>
>
>  --- Organization: We have made an effort to revise the paper and keep the discussion in the first section light and intuitive. Our model is quite technical, so a degree of formality is required in stating our main results in Section 2.1. We think it's a great suggestion to move Figure 2 forward to the introductory sections and have done so.

---

> ### Author Response · Authors · 2022-12-02
> **Response to Reviewer dBYC: Technical Novelty**
>
> Thank you again for your comments. We realize that your primary concern appears to be the technical novelty. We would like to take this opportunity to provide additional clarifications. (Some of these points might overlap with the points in our previous responses.)
>
> --- **Computational LB for SBM was not established in Chen and Xu (2014).**
>
> They only conjectured that there was such a computational LB but did not prove it.
>
> --- **Our computational LB is non-trivial because the planted clique reduction does not work.**
>
> For SBM, the only known computational LB is in Hajek, Wu and Xu (2015). They considered a 2-community SBM where the within-community probability, $p$, and between-community probability, $q$,  are in the same order. When $p$ and $q$ are in different orders, no computational LB has ever been established (even for SBM). The main challenge is that the common technique of reduction to a planted clique problem no longer works. It took us a long time to realize this issue and search for a new proof technique.
>
> --- **Our analysis of SgnQ test is more complicated than that in Jin, Ke and Luo (2021).**
>
> The SgnQ statistic decomposes into 256 different terms. The analysis requires to obtain a proper variance bound for each term. Our setting is different from the one in Jin, Ke and Luo (2021) in that the communities have severely unbalanced sizes and that the entries of the 2-by-2 matrix P are not at the same orders. Consequently, the variance bound for most of the 256 terms has changed. For example, one term is $X=\sum_{i,j,k,\ell} B_{ij} W_{jk}W_{k\ell}W_{\ell i}$, where $W$ is a generalized Wigner matrix and $B$ is a non-stochastic matrix. In Jin, Ke and Luo (2021), the variance calculation repeatedly used an inequality $\mathrm{Var}(W_{jk} )\leq C\theta_j \theta_k$. Unfortunately, it does not hold in our setting, hence, the variance bound of $X$ changes significantly. The case is similar for other terms. It took us a long time to calculate the new variance bounds for all 256 terms and figure out the correct detection boundary of SgnQ.

---

> > ### Comment · Reviewer_dBYC · 2022-12-06
> > **Thanks for answering the questions**
> >
> > Thank you the detailed answers to my questions. I have adjusted my score accordingly. Still, I would suggest improving the presentation by including some of these discussions, as it can be helpful to provide some motivation to readers.

---

> > > ### Author Response · Authors · 2022-12-06
> > > **Thanks for the reply**
> > >
> > > Thank you for the reply. We are grateful to you for considering our response and deciding to raise the score. In the final version we will be sure to include further discussion of the motivation and novelty of our work. We also welcome further suggestions for improvement and will include them in the next revision.

---

### Decision · Program_Chairs · 2023-01-20

**Decision:**

Accept: poster

**Justification For Why Not Higher Score:**

The paper is nice, but I did not identify any specific reason to highlight it.

**Justification For Why Not Lower Score:**

It could be rejected if the senior chair or the program committee considers it out of scope. It would indeed be more suitable for TCS or statistics venues.

**Metareview: Summary, Strengths And Weaknesses:**

The reviewers agree that the results of the paper are an interesting contribution to the statistics literature on community detection and its algorithmic hardness. The paper is of theoretical nature focusing on testing whether a small community is present of not the in degree-corrected SBM. This poses some mathematical challenges that are resolved in the paper. We were not sure ICRL is the best venue for this paper as the paper does not really deal with learning. But putting this aside it is a paper that should be accepted for publication.

**Note From Pc:**

if the above contains the word "oral" or "spotlight" please see: "oral" presentation means -> notable-top-5% and "spotlight" means -> notable-top-25%. As stated in our emails, we are disassociating presentation type from AC recommendations